# Central American and Caribbean tourism destinations' competitiveness: A temporal approach

**Víctor Ernesto Pérez León**[1]*, **Maria Amparo León Sánchez**[2], **Flor Mª Guerrero**[3]

**1** Faculty of Economics and Business Sciences, Department of Applied Economics II, University of Seville, Seville, Spain, **2** Faculty of Technical Sciences, Mathematics Department, University of Pinar del Río, Pinar del Río, Cuba, **3** Faculty of Business Sciences, Economics, Quantitative Methods and Economic History Department, Pablo de Olavide University, Seville, Spain

* vpleon@us.es

**Data Availability Statement:** The data is available upon request to the World Travel and Tourism Council in the following link (https://wttc.org/Research/Data-Enquiries-Form). Please, indicate the period 2000 to 2019 and the values of the

## Abstract

This study presents a new method for measuring tourism destination competitiveness based on data covering a specific time span. Issues, such as the type of data gathered, tools and methods employed, and the size and number of destinations, are addressed, as is the consideration of a specified time period. The proposal is based on the information given by linear regression equations, which not only enables the behaviour of destinations to be observed over time, but also facilitates their comparison. The data employed was from the period 2000–2019. Cluster Analysis was introduced to group destinations according to their performance. Moreover, various aggregation methods are proposed to obtain competitiveness rankings. A comparison between destinations was carried out using the non-aggregative and an aggregative approach. Certain destinations attained better positions than others that are considered as being more competitive in global international rankings. Five clusters were clearly identified. The results were consistent with the World Travel and Tourism Council outputs and underlined the importance attached to tourism development in the destinations from Central America and the Caribbean.

## Introduction

In order to contribute towards their economic and social development, several countries are improving their tourism sectors, thereby increasing the number of tourism destinations worldwide. At the same time, the quantity of originating markets has also augmented, albeit to a lesser degree. This asymmetry has led to fierce competition in the international tourism market, which is constantly on the rise [1]. As a result, tourism destination competitiveness (TDC) has emerged as a pre-eminent concept for policymakers and scholars and, therefore, its evaluation has gained special attention in the recent literature [2].

Several authors have striven to evaluate the competitiveness of tourist destinations over the years [3–6]. The measurement and comparison of tourism competitiveness is not an easy task, and despite the efforts registered in the literature, the results thereof measurement of

following indicators in percentage, available in the Economic Impact of each country: 1. Travel & Tourism Direct Contribution to Gross Domestic Product. 2. Travel & Tourism Total Contribution to Gross Domestic Product. 3. Travel & Tourism Direct Contribution to Employment. 4. Travel & Tourism Total Contribution to Employment. 5. Domestic Travel & Tourism Spending. 6. Leisure Travel & Tourism Spending. 7. Business Travel & Tourism Spending. 8. Visitor Exports. 9. Internal Travel & Tourism Consumption. 10. Government Individual Travel & Tourism Spending. 11. Capital Investment. 12. Outbound Expenditure.

**Funding:** P. V. E. Grant Number: RES/19/12/2017, Asociación Universitaria Iberoamericana de Postgrado (AUIP) P. V. E. Grant Numbers: P18-RT-1566, UMA18-FEDERJA-065, Agencia de Innovación y Desarrollo de Andalucía L. M. A. Grant Number: PI-195, University of Pinar Del Río, Cuba https://www.upr.edu.cu F. G. Grant Number: SEJ332. Secretaría General de Universidades, Investigación y Tecnología, Junta de Andalucía.

**Competing interests:** The authors have declared that no competing interests exist.

destination competitiveness remain tenuous [7]. However, the progress presented to date reveals several limitations regarding the selection of evaluation variables, the calculation of their respective weights [8], and the aggregation procedure used. Furthermore, we would like to point out that the temporal aspect constitutes a further issue regarding the evaluation of TDC.

Diverse studies evaluate TDC at a given moment in time [9–11], which is a widely used and convenient approach. This enables the competitive position of a destination with respect to its main competitors to be assessed at a certain moment, whereby the competitiveness is evaluated from a static point of view. As a result, the competitive position of a destination is evaluated by considering the values of the indicators at a given moment and, therefore, it is impossible to evaluate whether the policies and decisions made have contributed towards the improvement of the competitiveness along time.

Under this approach, the level of competitiveness of a certain destination may be affected by external factors in such a way that other destinations could be considered more competitive, not from improving the values of their indicators, but instead due to the deterioration of this first destination. In this respect, the use of a dynamic approach, which permits competitiveness to be analysed over a period of time, enables a destination to be analysed as to whether it is capable of improving its levels of competitiveness over time, in such a way that a higher value of competitiveness is associated to increasing behaviour over time.

Within the static approach, a latent debate rages on regarding which is the most feasible methodology to create TDC rankings [12]. However, despite advances in the area [9–11], differences between the proposed procedures remain, and some of the proposed alternatives imply the utilisation of algorithms that can, on occasions, create measures of tourism competitiveness that are difficult to explain to decision-makers. This seriously reduces their usefulness to end users.

Other researchers evaluated the TDC within a time span [13–16]. Most of these studies are more focused on the determination of the factors that influence the competitiveness of the destinations than on the analysis of the competitiveness positions of the destinations. Some researchers suggest segmenting the sample based on the destination characteristics and then analysing these segments based on smaller sub-samples of similar destinations [14]. However, these proposals continue to seek a better way to employ all the information. In this respect, the present research aims to contribute towards the literature in seeking methods to measure TDC by addressing several of the aforementioned gaps in research. First, the proposal involves measuring the TDC over a period of time, thereby analysing it from a dynamic approach.

In order to make a suitable diagnosis of competitiveness in this sector, it is important in the analysis to take into consideration the period in which impacts occur [13]. The proposal involves using the slope of the regression equation for each indicator in each destination, based on ordinary least squares (OLS). This enables the average behaviour of a destination within a time span to be identified, thereby leading to a better competitiveness position. As a result, competitiveness can be analysed as a dynamic and not as a static approach. The proposal permits the inclusion of all information available in each indicator in such a way that a destination's competitiveness is not only affected by the initial and final values, but also by all the intermediate values within the time span. Consequently, it will be possible to analyse whether the administrative decisions made over time contribute towards the improvement of the behaviour of the destinations, measured by the average behaviour of the indicator values.

The proposal determines the way in which destinations improve their TDC during a time span with respect to themselves [1]; that is, whether each destination manages to achieve, to a greater or lesser extent, its objectives, in accordance with its development possibilities and economic conditions. Moreover, the study allows the way in which the destinations improve their

TDC with respect to their competitors to be investigated [10,17], thereby taking into account the importance that comparison with other competitors holds in the measurement of TDC. Additionally, Cluster Analysis is proposed in order to group destinations according to their behaviour in the indicators under study.

A new way to analyse the initial information is presented. The proposed method is easy to apply, and the results are comprehensible and, therefore, easy to interpret for the end users. Moreover, the results can be analysed using either the non-aggregative or the aggregative approach. This latter approach involves the creation of a composite index. To this end, various methods are also proposed to take advantage of the information covering the whole time span.

This allows us to investigate a way to measure competitiveness over a given period of time, in such a way that it is seen as a dynamic and not a static process, whose results are easily understood by decision-makers. This process enables the behaviour of the level of competitiveness of destinations to be analysed visually. Furthermore, the proposal permits competitiveness to be analysed over a period of time without the need to carry out the calculation procedure for each year or sub-period analysed, thereby reducing the computational cost.

Prior studies conclude that competition occurs on multiple levels. [4] present a detailed list of applications for a diverse size of destinations. In this study, the destinations are the countries, whereby their facility to provide the information is considered. This helps towards the better provision of data when diverse years are included in the analysis. As a consequence, a further innovation of the study is that of the sample. This is a unique dataset of 33 destinations from the Caribbean region, which is, to the best of our knowledge, one of the largest samples of this kind in the studies developed in the region that involves destinations and number of years. Thirty-three tourism destinations between island states and continental states from Central America and the Caribbean are included. The selection of the competitors responds to the criteria of geographical proximity and market share. Furthermore, this selection is also due to the similarity of the destinations' tourism development. These countries compete for tourism mostly from the USA and Europe. This renders them potential competitors, typically in cruise tourism, which is strong in the region.

Moreover, only 17 from the 33 destinations incorporated in the study have been considered at least once in the Travel and Tourism Competitiveness Index of the World Economic Forum [6], which is the most highly recognised TDC raking from the managerial and research point of view. It is considered to be the best existing index in terms of comprehensiveness and methodological development at international level [15]. This index embraces a high number of destinations and indicators and serves as a guide for policymakers, tourism managers, and researchers alike [2,3,9]. However, it requires a greater amount of information, which is not easily attained for many countries, and these are therefore excluded from the ranking, such as various destinations in the Caribbean region, which have been excluded from the editions from 2017 and 2019 due to information unavailability, despite being the most tourism-dependent region worldwide [18].

Given the objective of this research, the data used for the study corresponds to 12 variables registered by the World Travel & Tourism Council (WTTC) for all destinations that are representative of the tourism industry in the period from 2000 to 2019. The proposal will provide researchers, policymakers, and stakeholders with the means to identify the key policies to stimulate the competitiveness of the destinations considered.

## Measurement of tourism destination competitiveness

One of the key issues in tourism competitiveness involves its means of measurement. With wide variability in the number of determinants employed to measure tourism destination

competitiveness, greater complexity and extensity can be observed in terms of the list of indicators utilised to measure each determinant, with the total number of indicators varying from one study to another [19]. There is a plethora of research that uses a different number of indicators to measure TDC [6,9,11,14,16,20,21]. Unfortunately, no consensus exists on which measurement best represents tourism competitiveness since each method has both strengths and weaknesses [7] and the number of indicators depends on the researcher's intention and the information availability.

For instance, in TDC studies that demand a high number of indicators, most developed countries succeed in collecting reliable tourism data, while less developed countries struggle to provide accurate and timely statistics [12]. Said countries therefore tend to be fewer or to be dismissed, as is the case of 15 destinations of the sample which have yet to be considered in the WEF's Travel and Tourism Competitiveness Index (TTCI), despite the importance of tourism therein and consequently the necessity for them to be compared to other destinations worldwide.

The debate in TDC measurement also involves the types of data gathered from the list of indicators used, whether they be hard or soft types of data [19]. There are studies that use soft data, collected mainly through surveys [22–25] Hard data, however, typically included in assessments of destination competitiveness, helps to conveniently gather large volumes of data and destinations [7,13,26], leads to more precise and accurate results, and is available over time, thereby facilitating the realisation of longitudinal studies. Hence, hard data is the type proposed in the present research.

Another matter concerning TDC measurement refers to the methods and tools used. There are several methodologies registered in the literature, each one featuring its own strengths and weaknesses. These are chosen according to the researcher's intentions. Principal Component Analysis (PCA) is one of the most commonly utilised for its ability to reduce information [2] as is Cluster Analysis [20]. Innovative Multi-Criteria Decision Methods (MCDM) can also be found [8,11], as can other applications, such as PROMETHEE [27], Goal Programming [5] and the combination of MDCM with others such as $DP_2$-Distance [10] and Data Envelopment Analysis [9].

Furthermore, Importance Performance Analysis (IPA) has recently gained popularity in the field [3,19]. Moreover, Structural equation modelling should be mentioned [28] and Regression methods [13,15,16], which are used both to forecast and to determine the importance of the determinants of the competitiveness [25]. In general, it should be stated that there is no methodology designated as the best to measure TDC. The choice of methodology involves the decision-maker's preferences and depends on its ability to analyse the results obtained. The present study aims to propose a method that uses the information gathered from the Linear Regression Equation as the initial data to measure TDC over a period.

The size of destinations is another topic concerning tourism competitiveness studies. This has been addressed at several levels (i.e., firm, regional, and national levels) [7]; and has included resorts [24] regional locations within the same country [8,27,29,30] or destinations from different countries [3], tour operator and hotel companies [31], cities [32], municipalities [2], regions [1,26] and countries [4,6,10,13,20]. Given the aim of the present research to include countries from the Caribbean region in TDC studies and also to consider the aforementioned availability of information, the selected destination size for the study is that of the country level. This is consistent with the view of several of the aforementioned applications.

The number of destinations constitutes another subject. This depends on the range of the study, the stakeholders' necessities, and on which places are considered to be the destination's principal competitors. There are studies focused on a single destination [1,33] in which the researchers strive to determine whether it is competitive or not, or to test a new TDC model.

This quantity may increase up to three destinations [34], up to ten [35,36], or even more [2], depending on the researchers' intentions. Furthermore, the number of destinations can be conditioned by information availability, which is closely related to the destination's economic conditions. Studies spanning a greater number of destinations are mostly associated with TDC rankings [5,9,10,21]. As can be observed, there are several proposals which differ according to the procedures utilised to create the indices, such as the weighting method of the indicators involved, the aggregation procedure, and the ease in interpreting the results. Hence, new approaches strive to guarantee a high explanatory power.

Additionally, the authors would like to highlight a feature related to the time span for which the TDC is analysed. Most studies use data from a single moment. This implies obtaining a single measure of each indicator [8,11,21] TDC is viewed as a static phenomenon because its value represents the state of the destination at a specific moment. This approach is useful in the evaluation of destinations and in their comparison with each other. However, it cannot be used to analyse the performance of a destination over time, unless the same measurement is carried out with the same set of indicators at another moment.

Very few studies use data covering a period [14,16] and incorporate measures for the same indicators across a time span [13] in such a way that it is not simply the direct value of the indicators that is analysed, but their evolution over a period. The use of information from a time span enables the evaluation of the tendency of the indicators over time. Consequently, indicator growth rates are approximate to the concept of competitiveness because they indicate the change of those levels acquired over time [13]. This view permits the competitiveness of the sector to be observed by analysing the period in which the impacts occur.

As a result, based on this approach, a destination's competitiveness can be observed through a single value for which a single calculus operation is required, and it is not necessary to perform as many calculations as intervals included in the period. In terms of TDC, policy makers must determine whether a destination performs better than do its core competitors at a given moment. Moreover, it can be observed whether the self-improvement or failure rate of a destination over time can be identified. In such a way, TDC is viewed as a dynamic phenomenon, which is consistent with its nature.

Since, for most TDC studies, the competitiveness level of a destination is associated with the way in which it achieves high scores for the indicators observed, this research aims to analyse the way in which a destination improves indicator values towards a better position of competitiveness. To this end, destinations from Central America and the Caribbean are studied based on the information of a group of indicators for the period 2000–2019, and their improvement or decreasing rate is determined in such a way that all the available information is taken into account. A destination's competitiveness can therefore be based on the indicators' rate of growth rather than on its levels.

The proposed methodology is based on linear regression equations. These are both easy to obtain and comprehensible. Furthermore, linear regression is available in most statistical software and it considers all values for every indicator in each year included in the data and hence no information is lost. The value obtained considers all the data available. Outcomes are affected positively or negatively for good or bad indicator values respectively, which is a desirable characteristic in measurement processes. The results are realistic and represent the average rate at which a given destination improves or declines regarding a specific indicator over the period considered. As a result, competitiveness is viewed as a dynamic process.

The number of destinations and their sizes pose no difficulty for the proposal. First, all the destinations deemed to be competitors may be included. The scores enable them to be ranked according to their own behaviour in each indicator. Along these lines, the sizes are not an issue because the measurement process demonstrates the behaviour of the destination in each

indicator with respect to itself across a time span. The mainstream TDC studies assume that one destination is more competitive than another if it obtains a better value for most of the indicators measured, which is a real assumption. This proposal additionally implies that a given destination could be considered more competitive if it is able to improve its indicator values with respect to itself more than do others over time. Alternatively, possible aggregation procedures are presented in order to create rankings using the average behaviour of the indicators as initial information.

## Economic impact of tourism in the region

Twenty-one (63.36%) of the countries involved are island states and 12 (36.64%) are continental states. Except for Puerto Rico, all these countries are associated with or members of the Caribbean States Association (CSA); 22 belong to the Caribbean Tourism Organization (CTO) and, according to the World Tourism Organization (UNWTO) classification, all 33 belong to the Caribbean or Central American region, except Mexico (North America) and Guyana, Suriname, and Venezuela (South America). For the purpose of this study, all these countries will be grouped into the same region. Other small islands and continental states of the area were not considered due to the lack of information. In general, for most of these countries, tourism constitutes their main source of income.

The 33 countries included in the study account for 33% to 36% of the international visitors to the continent between 2000 and 2019, thereby acting as the second-most visited region within the American continent. According to the World Tourism Organization information (UNWTO), this region maintained growth in its number of visitors during the period. It was less affected by the decline of the number of visitors in 2009. As a result, it can be assumed that this group of countries enjoyed greater tourist affluence in the time registered as the most critical during the period analysed. Moreover, starting from 2013, this group of countries had a tendency to increase more noticeably than North America, in spite of welcoming fewer tourists [37–39]. This confirms that the growth in the number of visitors in the Americas is due to the countries under analysis. Despite the augmentation of international tourist arrivals to the continent, this sub-region achieved the highest increment ratio.

Travel and Tourism growth in the Caribbean region was robust at 3.4% in 2019, as countries continued to recover from the 2017 hurricane season. Several countries underwent impressive growth in 2019 with Dominica increasing by 43.6%, followed by Anguilla with 19%. Other countries in the region demonstrated strong performances including St Kitts and Nevis (14.6%), Puerto Rico (10.1%), Barbados (9.7%), and St Vincent and the Grenadines (9%) [18]. An analysis of the most important origin markets shows that the increase in visitors was driven by North America, with travel from the USA (which accounts for 53% of visitors) up by 6.5%, and travel from Canada up by 12.2%.

The top Caribbean destination by far is the Dominican Republic, with a 29% share of visitors, followed by Jamaica, with 12%, Cuba with 11%, and the Bahamas with 7% [40]. The 38.2 million international visitors who visited the region in 2019 spent approximately US$35.1 billion and the sector represents 13.9% of the Gross Domestic Product (GDP) of the whole economy in the region. This is the highest contribution of tourism to any region's GDP registered worldwide in 2019 [41], 1thereby rendering it the most tourism-dependent region.

In addition, tourism is responsible for over 20% of the region's exports and 13.5% of employment. However, in many Caribbean countries, the sector accounts for over 25% of their GDP, which is more than double the global average of 10.4% [42]. Six of the destinations from the region are ranked among the Top 20 destinations worldwide with the fastest-growing contribution from the travel and tourism industry to their GDP [18] although not one of these

was included in the TTCI. Furthermore, from 2013, the region has been experiencing an increasing trend which is more noticeable than in North America, in spite of receiving fewer tourists. This confirms that the growth in the number of visitors to the Americas is thanks to the countries under analysis. Despite the augmentation of international tourist arrivals to the continent, the Caribbean achieved the highest increment ratio according to information from the UNWTO [37–39].

According to WTTC data, 11 countries from the region were recorded as ranking among the top 35 most tourist-dependent countries worldwide, measured by the contribution of the tourism industry to their GDP [43], with three more countries of the neighbouring area included within this sample of 35. Furthermore, 11 of these countries have featured at least once among the top ten worldwide regarding the issues measured by the WTTC, which is a demonstration of the dependence of this region on tourism.

With regard to the possibilities of tourist emissions of the area, a short analysis reveals that Caribbean inhabitants seldom frequent the zone as tourists. For Mexico and Central American countries, Caribbean visitors failed to reach 1%, while for South American countries, they represent 3.91%. Generally, tourists from the Caribbean prefer and are more likely to be tourists in their own area, rather than in the rest of the continent. A glance at the main tourism receptors worldwide reveals that the presence of Caribbean tourists is small, which underlines the lack of travel opportunities open to the inhabitants of the region and justifies their greater presence as tourists in their own territory. In addition, this highlights the weaknesses that they present in terms of economic development with respect to the remaining countries in Continent, which explains their scarce presence among global tourism providers.

## Methodology

### Data

Research that analyses TDC across a time span uses only a few indicators (usually no more than ten) [13,14,16]. However, this study involves 12 indicators used by the World Travel and Tourism Council (WTTC) to evaluate the result of the tourist activity at each destination through the contribution of tourism to GDP, plus Direct Spending Impacts and Indirect and Induced Impacts. The information covers the period from the year 2000 to 2019. The selection of indicators was conducted based on fundamental principles, including their relevance, analytical soundness, and accessibility of data [14]. This selection helps to prevent difficulties in obtaining reliable values for the indicators in developing countries, as recognised in the literature [12].

Information was available for all destinations, given that only 16 are included in all the editions of the TTCI from the list of 33 countries in the study. The indicators selected correspond to hard data. This enables a broader number of destinations to be included in the study. The indicators are listed below, and their values are presented in the local currency, in billions of US$, and as a percentage. This information is available on the WTTC DATA GATEWAY and in each country's report.

1. (GDP_DC) Travel & Tourism Direct Contribution to Gross Domestic Product: GDP generated by industries that deal directly with tourists, including hotels, travel agents, airlines and other passenger transportation services, and by the activities of restaurant and leisure industries that deal directly with tourists.

2. (GDP_TC) Travel & Tourism Total Contribution to Gross Domestic Product: GDP generated by direct Travel & Tourism industries plus the indirect and induced contributions, including the contribution of capital investment spending.

3. (EDC) Travel & Tourism Direct Contribution to Employment: The number of direct jobs within the Travel & Tourism industries. This includes employment by hotels, travel agents, airlines, and other passenger transportation services (excluding commuter services). It also includes, for example, the activities of the restaurant and leisure industries directly supported by tourists.

4. (ETC) Travel & Tourism Total Contribution to Employment: The number of jobs generated directly in the Travel & Tourism industry plus the indirect and induced contributions.

5. (DTTS) Domestic Travel & Tourism Spending: Spending within a country by that country's residents for both business and leisure trips. Multi-use consumer durables are not included since they are not purchased solely for tourism purposes.

6. (LTTS) Leisure Travel & Tourism Spending: Spending on leisure travel within a country by residents and international visitors.

7. (BTTS) Business Travel & Tourism Spending: Spending on business travel within a country by residents and international visitors.

8. (VE) Visitor Exports: Spending within the country by international tourists for both business and leisure trips, including transportation spending.

9. (ITTC) Internal Travel & Tourism Consumption: Total revenue generated within a country by industries that deal directly with tourists including visitor exports, domestic spending, and individual government spending. This does not include spending abroad by residents.

10. (GI) Government Individual Travel & Tourism Spending: Government spending on individual non-market services for which beneficiaries can be separately identified. These social transfers are directly comparable to consumer spending and, in certain cases, may represent public provision of consumer services. For example, it includes the provision of national parks and museums.

11. (CI) Capital Investment: Capital investment spending by all sectors directly involved in the Travel & Tourism industry. This also includes investment spending by other industries on specific tourism assets, such as new visitor accommodation, passenger transportation equipment, and restaurants and leisure facilities for specific tourism use.

12. (OTS) Outbound Expenditure: Spending outside the country by residents on all trips abroad. This expenditure occurs almost exclusively by resident visitors outside the economic territory or on trips to leave this economic territory (e.g., using a non-resident carrier).

As is expressed by the WTTC in each country's Economy Impact, the percentage of the total refers to each indicator's share of the whole relevant economy indicator such as the GDP for indicators 1, 2, 5, 6, 7, 9, and 12. For indicators 3 and 4, the percentage is with respect to the employment of the whole economy. The percentage of Visitor Exports (8) is relative to the total exports of goods and services. Government Individual Travel & Tourism Spending (10) is relative to the total tourism expenditure. Finally, Capital Investment (11) is relative to the investment of the whole economy. The fact that not all the indicators are evaluated with respect to the destinations' GDP is a desirable characteristic. This defends against the possibility of an increase in the contribution of tourism to the GDP of any indicator being caused by the decline of other activities instead of a development at a tourist destination.

Given the potential importance and contribution of tourism to a country's GDP, and of its benefits to a wide range of economic activities in the context of increased global competition,

tourist destinations have been forced to seek new ways to obtain a competitive advantage [6]. Indicators GDP_DC and GDP_TC quantify the relative importance of the tourism industry in each destination, and are valid for measurement of TDC [44]. According to [17], an improvement in Travel & Tourism competitiveness is an encouraging trend given that, in over half the countries in the Americas, the Travel & Tourism industry's share of GDP is greater than the aggregate global level. Those indicators that consider employment in the sector (EDC and ETC) are of major importance since competitive destinations provide and increase employment and value added by the tourism industry [45]. These indicators have been used in dimensions aimed at monitoring the evolution of destination competitiveness [20].

The development of tourism contributes positively towards the economic prosperity of countries for which the bidirectional causal relationship can be emphasised [20,46]. Tourism spending is undoubtedly a major key factor not only in terms of economic growth but also in terms of competitiveness [6]. The indicators concerning tourism spending therefore remain useful in measuring TDC. This is the case of DTTS, LTTS, BTTS, and ITTC used by [20] to evaluate competitiveness, and of the indicator VE, also used by [17] to analyse destination competitiveness, and of Outbound Tourism (OTS) [7].

The extent to which the government prioritises the Travel and Tourism sector exerts a significant impact on its competitiveness [6]. In this respect, national and local governments play vital roles in tourism development [5,6]. The indicator GI is representative of the efforts of governments to encourage tourism development. From among the indicators used in the study, this is the only indicator that coincides with those proposed by the WEF to create the TTCI [20]. This indicator has also been used in other studies aimed at measuring TDC [16]. Moreover, Capital Investment (CI) may be viewed as a contribution to local economies to invest in the tourism sector, whose objective is to lead to economic benefits.

## Linear regression model

The proposed approach for the evaluation of TDC is that of Linear Regression Modelling. This has previously been employed in tourism studies to forecast tourism demand [16] and to explain TDC determinants [15]. The main objective of this approach is to determine the degree to which one or more variables (independent) affect the dependent variable, and it presents diverse advantages [47]. The proposed methodology involves estimating the linear regression equation of each indicator for each destination. The indicators are the dependent variables, measured in percentages, while the independent variable is time (in years). All the dependent variables are expressed on a 0–100 scale, and therefore no normalisation process is required. The intercept represents the indicators' average value at time zero, that is, at the beginning of the period. The slopes of linear regression equations represent the mean annual performance rate of each indicator in the period with respect to the reference economic value, as is stated in the explanation of each indicator.

The use of these values as initial information enables the researchers to study the destination's behaviour over a period without the need to repeat the calculus for each year. It is possible to compare destinations by considering their self-performance for each indicator over time. Furthermore, slope values enable destinations to be compared according to their average growth rate owing to their unit invariance. This method has been proposed thanks to the possibility of analysing the behaviour of a destination while taking into consideration all the values of each indicator in the period. All the available information is therefore considered. A variation rate between two periods (initial and final) could alternatively be used but this would only consider the initial and final values, thereby missing the effect of the intermediate values throughout the time span.

The approach enables the behaviour of a destination to be observed by means of a single value for each indicator in a given time span. This is another way to observe the competitiveness of a destination. A destination may improve its level of competitiveness over time if it achieves a growth rate in its indicator values. The main objective is not to forecast, but instead to observe the average performance of the destinations in each indicator. This is the main advantage of the procedure applied with respect to the other approaches. Moreover, the slope is used as a unique measurement unit for all the destinations to determine groups with a similar growth rate in their indicators. The use of other methods could provide a better fit but, in certain cases, these would be different for each relationship. The use of various methods may cause an incomparability between destinations.

A negative value for the slope for the $i$th indicator of the $j$th destination indicates a negative linear association. This means that the value of the indicator decreases over time at a rate equivalent to the slope (Fig 1A). On the other hand, a positive score demonstrates an annual improvement ratio equivalent to the slope (Fig 1B). There are other issues, such as the absence of association and other strong relations, but not linear. For two given destinations ($i$ and $k$) evaluated in a certain indicator $\theta$, if $\beta_{0i} > \beta_{0k}$, then destination $i$ has a higher initial expected value than destination $k$ at time 0, that is, in the initial year. If, for the whole period, $\beta_{1k} > \beta_{1i}$, then destination $k$ has a better average growth rate than destination $i$. As a result, if both destinations maintain the same slope ($\beta_1$), then destination $k$ is going to achieve a higher value than destination $i$ for the indicator and $\theta_k > \theta_i$ as variable $x$ increases, that is, over time.

## Cluster analysis

The use of cluster analysis is proposed in order to observe how destinations can be clustered according to their behaviour in the period and to identify common patterns. This approach has been used in tourism studies due to its ease in identifying a group of units with similar characteristics according to the phenomena measured [48]. This is a multivariate method with the primary purpose of grouping. It is a very common statistical technique in which a set of objects (e.g., events, people) is subdivided into groups (clusters) in such a way that objects in the same group are more similar (based on certain variables) to each other than to those belonging to other groups [49].

A hierarchical cluster analysis with the Ward's method criterion was applied. This method was used for its ability to minimise differences within clusters. All the variables therein

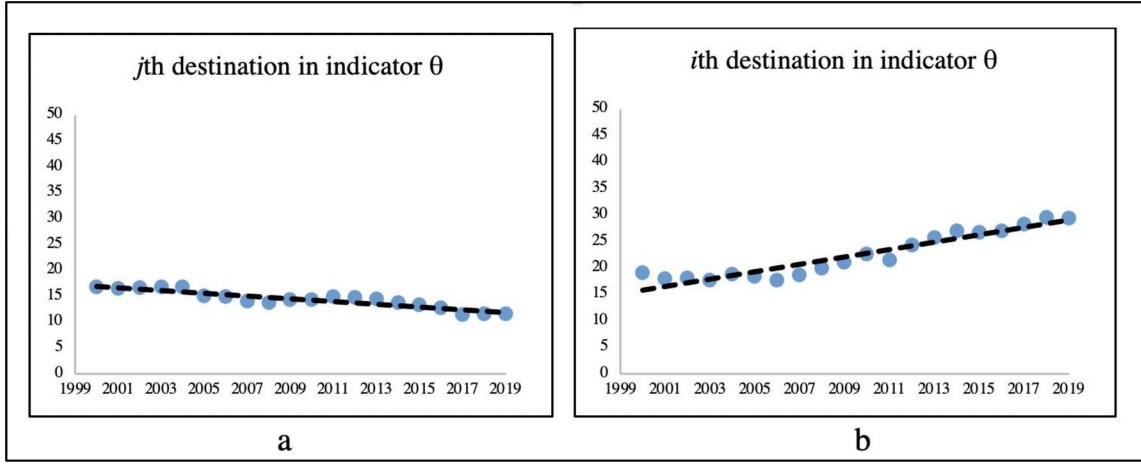

**Fig 1. Negative and positive linear regression.**

intervene to determine the distance between clusters. Furthermore, the sum of squares within the cluster is minimised in each step of the clustering process [50]. The squared Euclidean distance was used as a measurement, as is suggested in the specialised bibliography when the Ward method is used. It is common to find applications in which the arithmetic mean or the median is used to compare clusters with variables with repeated measures over time. However, the slope is considered a better option for the measurement of the average behaviour in data with a tendency.

## Results and discussion

In the following subsections, the results of model estimation, diagnostic indices, and ranking of the destinations are reported and discussed. The study of competitiveness is to be carried out based on the slope of the regression equation for each indicator and destination. The growth rates of the indicators are more approximate to the concept of competitiveness because they indicate the change in the time of those acquired levels [13]. The analysis will be made using both the non-aggregative and the aggregative approaches.

Perhaps not all destinations follow a linear trend in each indicator because of tourist flows. However, due to the necessity of establishing a comparison, it is not possible to use the model that best matches the behaviour of each destination in each separate indicator, but instead the model that is applicable to all the models and that provides an explanation of the results. To this end, linear regression could be considered one of most appropriate approaches thanks to its characteristics.

No other models were selected, since the aim is to demonstrate the behaviour of the indicators over time, thereby guaranteeing the greater explanatory power of the results. To this end, the linear model is the most suitable. The indicators, valued in percentages, were employed in order to preclude scale differences and to eliminate variable transformation. If data transformation had been necessary, then it would have affected the explanatory power of the score used. Furthermore, the remaining scales in which the data was offered in the WTTC (e.g., in the local currency and in billions of US$ in Nominal and Real prices) contribute nothing towards the differentiation of the destination, due to value similarity caused by data approximation. From an initial set of 7,926 values that resulted in 12 matrices of order 20 x 33, we now have only 396 values. These values are the slopes of the regression equations of each destination of each indicator for the period 2000–2019.

### Non-aggregative approach

The results are presented in Table 1, where the slopes of the regression equations of each indicator for each destination are shown. These values represent the annual mean variation for each destination in each indicator during the period 2000–2019, and incorporate the information of the values for the complete time span. As a result, the value is affected for every year included. The slopes act as the initial information for each destination.

Under this approach, if a destination performs equal to or better than that expected, then a positive competitiveness diagnosis is established, and a negative competitiveness diagnosis is established if the flow performs worse than predicted, as is stated by [8]. As a result, positive values demonstrate that a given destination maintains improvement towards a more competitive position with respect to its own planification. A growing tendency reveals the positive impact of the policies and decisions made whose objective is to develop the tourism sector. Following this explanation, a destination with a slope greater than others in a certain indicator demonstrates improved competitiveness. Consequently, it is possible to evaluate the level of competitiveness of a destination with respect to itself and with respect to the other competitors.

**Table 1. Slopes of the destinations.**

| Destinations | GDP_DC | GDP_TC | EDC | ETC | DTTC | LTTS | BTTS | VE | ITTC | GI | CI | OTTS | Ranking |
|---|---|---|---|---|---|---|---|---|---|---|---|---|---|
| Anguilla | 0.122 | 0.235 | -0.246 | -0.757 | 0.081 | 0.137 | -0.018 | -0.758 | 0.246 | 0.042 | -0.156 | -0.074 | 28 |
| Antigua & Barbuda | -0.271 | -0.526 | -0.271 | -0.526 | 0.056 | -0.267 | -0.005 | -0.559 | -0.465 | 0.110 | 1.219 | 0.057 | 32 |
| Aruba | 0.752 | 2.507 | 0.755 | 2.248 | 0.165 | 0.696 | 0.052 | 1.471 | 0.589 | 0.086 | 0.764 | 0.218 | 1 |
| Bahamas | 0.157 | 0.386 | -0.238 | -0.125 | 0.040 | 0.169 | -0.015 | 0.472 | 0.075 | -0.004 | 0.597 | -0.021 | 16 |
| Barbados | -0.035 | -0.152 | -0.092 | -0.315 | -0.006 | -0.050 | 0.015 | -0.016 | -0.066 | 0.041 | 0.427 | -0.151 | 25 |
| Belize | 0.362 | 1.157 | 0.314 | 1.037 | 0.042 | 0.322 | 0.038 | 0.736 | 0.374 | 0.051 | 1.107 | -0.132 | 13 |
| Bermuda | -0.013 | 0.247 | -0.064 | 0.156 | 0.230 | -0.023 | 0.010 | -0.285 | 0.106 | 0.024 | 0.187 | -0.138 | 19 |
| British Virgin Islands | 0.462 | 1.234 | -0.615 | -1.271 | 0.042 | 0.450 | 0.013 | 1.037 | 0.578 | 0.101 | 0.354 | 0.302 | 27 |
| Cayman Islands | -0.156 | -0.377 | -0.180 | -0.458 | 0.041 | -0.165 | 0.007 | -0.569 | -0.226 | 0.108 | 0.398 | 0.237 | 30 |
| Colombia | -0.017 | -0.051 | -0.015 | -0.040 | -0.032 | -0.006 | -0.011 | 0.241 | -0.003 | 0.003 | -0.116 | 0.013 | 14 |
| Costa Rica | -0.113 | -0.215 | -0.055 | -0.131 | 0.009 | -0.088 | -0.027 | -0.264 | -0.118 | 0.020 | -0.022 | -0.089 | 26 |
| Cuba | -0.060 | -0.143 | -0.046 | -0.116 | -0.006 | -0.062 | 0.002 | -1.516 | -0.128 | 0.020 | 0.842 | -0.037 | 29 |
| Dominican Republic | 0.219 | 0.673 | 0.202 | 0.614 | -0.031 | 0.187 | 0.030 | 0.741 | 0.362 | 0.015 | 0.205 | -0.023 | 8 |
| Dominica | -0.038 | -0.096 | -0.071 | -0.164 | 0.029 | -0.042 | 0.003 | 0.254 | -0.028 | 0.074 | 0.084 | -0.033 | 17 |
| El Salvador | 0.080 | 0.219 | 0.077 | 0.207 | 0.096 | 0.062 | 0.017 | 0.288 | 0.118 | 0.011 | 0.238 | -0.023 | 6 |
| Grenada | 0.221 | 0.713 | 0.218 | 0.665 | -0.016 | 0.085 | 0.135 | 0.647 | 0.414 | 0.040 | 0.702 | -0.046 | 9 |
| Guadeloupe | -0.069 | -0.301 | -0.038 | -0.250 | -0.013 | -0.054 | -0.012 | -1.998 | -0.069 | 0.004 | -0.204 | -0.140 | 33 |
| Guatemala | -0.003 | 0.035 | 0.000 | 0.037 | -0.004 | 0.001 | -0.004 | 0.007 | -0.015 | 0.018 | 0.376 | -0.009 | 7 |
| Guyana | 0.000 | -0.012 | 0.004 | -0.008 | 0.010 | 0.019 | -0.020 | -0.174 | -0.054 | 0.027 | -0.352 | 0.011 | 22 |
| Haiti | 0.085 | 0.250 | 0.058 | 0.184 | -0.015 | 0.072 | 0.013 | 0.833 | 0.104 | 0.004 | 0.264 | 0.053 | 5 |
| Honduras | 0.057 | 0.216 | 0.050 | 0.189 | 0.075 | 0.021 | 0.036 | -0.021 | 0.032 | 0.015 | 0.332 | -0.055 | 10 |
| Jamaica | 0.093 | 0.394 | 0.078 | 0.339 | 0.081 | 0.039 | 0.054 | 0.838 | 0.154 | 0.047 | 0.461 | 0.056 | 2 |
| Martinique | 0.019 | 0.021 | 0.027 | 0.031 | -0.023 | 0.032 | -0.014 | -0.239 | 0.009 | 0.005 | -0.200 | -0.057 | 20 |
| Mexico | -0.010 | -0.018 | -0.113 | -0.201 | -0.041 | -0.012 | 0.001 | -0.053 | -0.074 | 0.030 | 0.067 | 0.013 | 18 |
| Nicaragua | 0.131 | 0.298 | 0.031 | 0.150 | 0.069 | 0.104 | 0.026 | 0.036 | 0.119 | 0.007 | 0.198 | 0.014 | 3 |
| Panama | 0.193 | 0.513 | 0.250 | 0.578 | -0.011 | 0.202 | -0.009 | 1.144 | 0.303 | 0.018 | -0.094 | -0.018 | 12 |
| Puerto Rico | 0.010 | 0.026 | -0.004 | -0.003 | -0.010 | 0.011 | -0.001 | 0.013 | 0.014 | 0.020 | 1.093 | -0.070 | 11 |
| St. Kitts & Nevis | -0.066 | 0.271 | -0.058 | 0.243 | 0.008 | -0.224 | 0.155 | -0.265 | -0.214 | 0.039 | 0.960 | 0.095 | 24 |
| St. Lucia | -0.025 | -0.131 | -0.025 | -0.131 | 0.069 | -0.076 | 0.049 | 0.125 | 0.045 | 0.028 | -0.209 | 0.016 | 21 |
| St. Vincent & the Grenadines | -0.297 | -0.603 | -0.236 | -0.495 | -0.064 | -0.247 | -0.053 | -0.256 | -0.523 | 0.032 | 0.493 | -0.006 | 31 |
| Suriname | -0.024 | -0.093 | -0.040 | -0.125 | -0.022 | -0.014 | -0.010 | -0.168 | -0.036 | 0.006 | -0.011 | -0.136 | 23 |
| Trinidad & Tobago | -0.012 | -0.019 | -0.037 | 0.023 | 0.150 | 0.019 | -0.031 | -0.129 | 0.029 | -0.007 | -0.031 | -0.072 | 15 |
| Venezuela | 0.022 | 0.075 | 0.005 | 0.021 | 0.049 | 0.034 | -0.012 | 0.088 | 0.018 | 0.007 | 0.152 | 0.043 | 4 |

The analysis can be developed by means of indicator, to identify regional trends or by destination, to evaluate the attainment of individual competitiveness. With the indicator approach, it is possible to observe the importance given to tourism in all the countries in the area. A general local government concern regarding the development and maintenance of non-market services whose beneficiaries can be both local and international tourists in the region is evident from the 31 positive values out of the potential 33 for slopes of Governmental Individual Travel & Tourism Spending. Directly related to this concern is a common regional interest, observed in the development of travel and tourism services. This is justified by the 23 countries of the sample with positive values for their slopes in Capital Investment. This is evidence of the importance given to this economic activity in the area, and is consistent with the relative best position of the region in this matter worldwide [51].

In contrast, despite the positive performance in GI and CI in the region, and the presence of more average yearly growth than average yearly decline, the most widespread local

difficulties are related to the capacity to create jobs in the tourism industry. The direct and total contribution to employment shows a decreasing behaviour in 19 and 18 of the 33 economies, respectively. This should be a general concern for these countries whose objective involves improving the standard of living of the local population. In spite of this almost generalised negative performance, this was the region for which the Travel and Tourism industry most contributed to the total employment in 2017 (in relative terms), according to [43]. Furthermore, in agreement with WTTC's forecast, this area is expected to achieve the highest relative growth in this indicator for 2028 [51].

However, the most concerning indicator is that related to the tourism emission of the region. This is consistent with the condition of these destinations as underdeveloped countries. A glance at the main tourism receptors worldwide reveals that the presence of Caribbean tourists is small, which 1underlines the lack of travel opportunities open to the inhabitants of the region. In addition, this highlights the weaknesses that they present in terms of economic development with respect to the remaining countries in the region, which explains their scarce presence among global tourism providers.

The analysis may be also carried out under the destination approach, to analyse individual behaviour. Extreme scores show that the highest values for all the indicators appear in only five destinations. These countries achieved the highest annual increase of all the issues analysed in the period. Aruba attains the best achievement in seven indicators: the two related to employment (EDC and ETC), also those representative of the contribution to the GDP (Direct and Total), that of tourism spending within the country (VE), and the ITTC and LTTS. The next best achiever is Antigua and Barbuda, with the highest yearly growth rates in the two indicators that represent the efforts of the government related to tourism development: Government Individual Travel & Tourism Spending (GI) and Capital investment (CI). These are followed by Bermuda, with the best increment in Domestic Travel & Tourism Spending (DTTS), the British Virgin Islands, with the best slope for the Outbound Tourism (OTS), and Saint Kitts and Nevis, with the best improvement in business trips in the region (BTTS).

All the worst performances are negative and distributed across seven destinations. Saint Vincent and the Grenadines has the highest decline in five aspects: BTTS, GDP_DC, GDP_TC, ITTC, and DTS. This seems to be the least competitive destination or, at least, that with the worst averages registered in the period, in a number of indicators. The British Virgin Islands has the worst performance in EDC and ETC. Subsequently there are Antigua and Barbuda, Barbados, Guadeloupe, Guyana, and Trinidad and Tobago, which achieved the worst performances in LTTS, OTS, VE, CI, and GI, respectively.

A detailed inspection reveals that only three destinations have improved in all the indicators for the period with respect to themselves. These are Aruba, Jamaica, and Nicaragua. Despite the great behaviour of these destinations in all the indicators considered, there exist others with average growth rates higher than these in their indicators. Consequently, in comparison to the other destinations, Aruba, Jamaica and Nicaragua have achieved a lower competitiveness position in certain indicators. As a result, a positive performance in all the indicators alone is insufficient; higher values than those of the remaining competitors is also necessary for a country to be considered the most competitive. Belize and El Salvador have an increased level in all indicators except one (OTS), while Haiti has the same behaviour except for negative values attained in the DTTS.

In contrast to previous studies [5,9–11], this approach involves the evaluation of competitiveness in a dynamic way. Mexico could be viewed as the most competitive destination of the sample. It is included among the top ten destinations worldwide for almost all the indicators measured, except for VE, BTTS, and CI. This is consistent with this country's size and its established tourism industry, which is of sufficient standing as to be ranked among the best

destinations according to the WTTC every year. However, this proposal observes how much a destination has been able to improve the topics analysed with respect to itself over a specific period of time.

As a result, Mexico remains far from being the destination with the greatest average growth rate, with only four indicators with positive behaviour over time, from the year 2000 to 2019. This means that, despite prevailing as one of the most competitive destinations worldwide, Mexico does not present positive behaviour throughout the period. However, the values of its indicators are so large in comparison with the destinations in the area that, even with a decreasing rate, their absolute values remain higher than the other destinations. As a consequence, our proposal allows us to observe how other destinations in the region have not only been able to increase the values of their indicators over time in a greater way than Mexico, but also with respect to themselves. It is therefore possible to identify how small destinations such as Aruba have attained higher growth rates than others that have been established as competitive destinations in the international tourist market for a long time [8].

An analysis between islands and continental states can also be carried out. Both groups have similar performances in GI and CI, consistent with the regional interest in supporting tourist services and products in favour of tourism development. Additionally, for most countries of both groups, 57.14% of island states and 58.33% of continental states, the total contribution of tourism to GDP (GDP_TC) has undergone annual growth in the period. This is more intense for island states. This issue situates these economies in a more tourism-dependent condition, while continental countries are in possession of sources other than tourism to support their economic development.

The stark difference between these groups is due to the Total Contribution of tourism to employment with a yearly increment of 28.6% for islands versus 66.7% for continental states. Moreover, island states have a year average growth that is higher than that of continental states in the investment in new visitor accommodation, passenger transportation equipment, and in restaurants and leisure facilities for specific tourism use (76.2% vs. 58.3%). This is consistent with the dependence on tourism registered by the Island States [18] and, therefore, underlines the necessity to improve their tourism potential.

## Results for cluster analysis

Cluster analysis involves seeking similarities in TDC in the destinations of the region. The results clearly identified five groups. The number of groups was decided based on the dendrogram information (Fig A.1, Appendix A in S1 Appendix) and the result of the F tests, which revealed great differences between groups, and on the Kruskal-Wallis test. The researchers' criteria regarding the explanation of group characteristics were also taken into account. The first two clusters are each formed of a single destination: Cluster 1, Aruba; Cluster 2, the British Virgin Islands. The remaining three clusters contain 7, 7, and 17 destinations, respectively.

The Kruskal-Wallis test demonstrates that 8 of the 12 variables considered reveal significant differences between groups (Table 2). The remaining 4 variables do not contribute towards the differentiation of the groups. Of these 4, the first is that of Government Travel & Tourism Spending (GI), for which most countries of the region attained positive behaviour in the period, consistent with the achievement of [20] in the measurement of TDC. Moreover, this is also associated to the outputs of [14] for the tourism-based countries that recognise the importance of developing resources that generate value for the tourism industry and for the broader economy.

The other two variables that do not contribute to differences between groups are Business Travel & Tourism Spending (BTTS) and Domestic Travel & Tourism Spending (DTTS), with

**Table 2. Kruskal-Wallis test.**

| | Test Statistics[a,b] | | | | | | | | | | | |
|---|---|---|---|---|---|---|---|---|---|---|---|---|
| | **BTTS** | **LTTS** | **E_TC** | **GDP_TC** | **ITTC** | **DTTS** | **GDP_DC** | **GI** | **CI** | **VE** | **E_DC** | **OTTS** |
| Chi-Square | 4.339 | 18.424 | 11.414 | 16.002 | 18.899 | 3.908 | 18.357 | 7.696 | 12.637 | 21.685 | 11.595 | 6.681 |
| df | 4 | 4 | 4 | 4 | 4 | 4 | 4 | 4 | 4 | 4 | 4 | 4 |
| AsympSig. | .362 | .001 | .022 | .003 | .001 | .419 | .001 | .103 | .013 | .000 | .021 | .154 |

a. Kruskal-Wallis Test.

b. Grouping Variable: Number of initial cases.

more than 50% of the destinations with positive behaviour, but only a small difference between maximum and minimum values. This is also related to the findings of [20] reveal the close relationship of the variables that measure tourism spending and competitiveness. In addition, the region is located in one of the last positions worldwide for these two items according to [51] which demonstrate low regional competitiveness. Finally, the prevalence of negative values for the Outbound Expenditure of the region, associated to the condition of underdeveloped countries, reports no differences. The Kruskal-Wallis test results and the WTTC outputs were consistent.

In the first two clusters, each formed by a single destination (Aruba and the British Virgin Islands), Aruba was the destination with steepest slopes in seven indicators. These indicators are comprised among the eight from the sample that causes significant differences between the clusters. Furthermore, Aruba attained the second highest value in one other indicator (DTTS) and presented no negative values. The British Virgin Islands provided the highest value only in Outbound tourism (OTS); however, this destination accounts for the second-highest values in four other indicators, which cause significant differences. Moreover, this destination registered the highest negative behaviour in the two indicators related to the contribution of tourism towards employment. These two countries are located among the top ten destinations worldwide with the best achievement levels in GDP_DC, GDP_TC, VE, LTTS, and CI, according to the WTTC.

Figs 2–4 demonstrate the behaviour of the slopes for each destination in each indicator in the period. The third cluster comprises five island states and two continental states (Fig 2). In general, these destinations have an average positive performance in almost all the indicators,

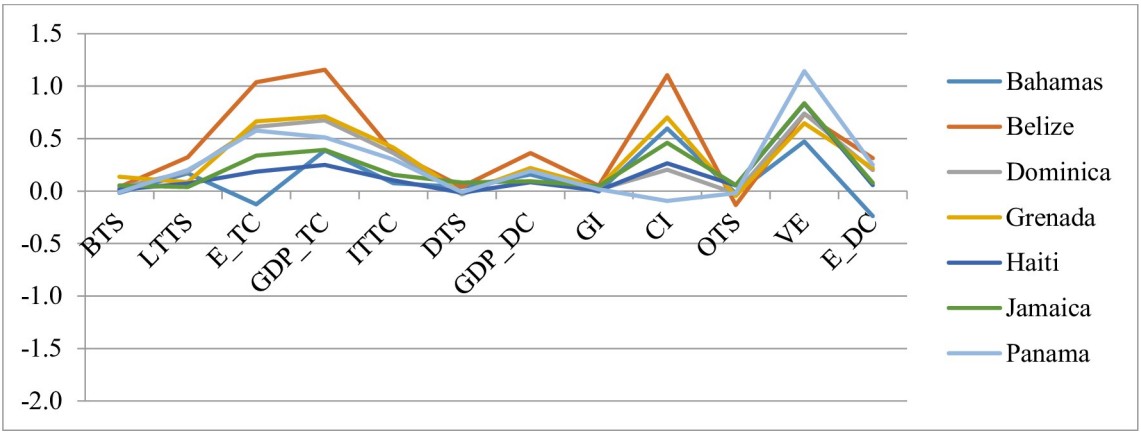

**Fig 2. Slopes for the 3rd cluster.**

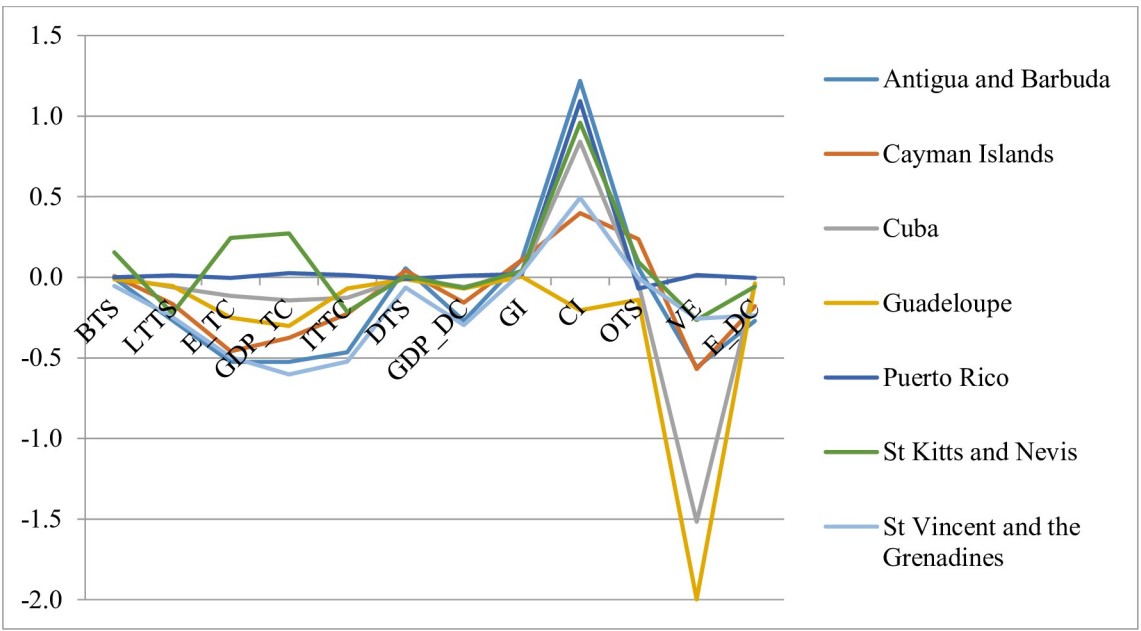

**Fig 3. Slopes for 4ᵗʰ cluster.**

with the exception of the outbound expenditure, which contributes no significant differences between groups. The seven destinations maintained a growing rate in LTTS, GDP_DC,

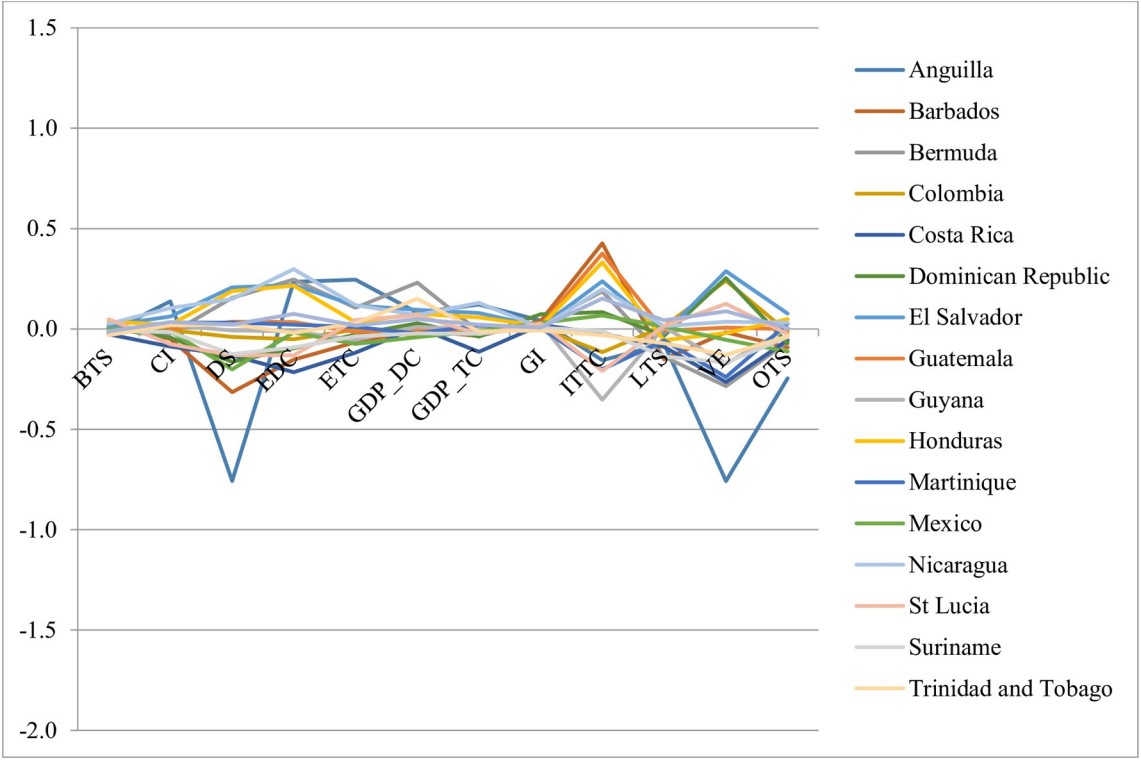

**Fig 4. Slopes for 5ᵗʰ cluster.**

GDP_TC, ITTC, and VE. These variables are all significant in showing differences between groups. Moreover, in another four indicators, a single destination attained a negative value, which demonstrates the overall positive behaviour of this cluster. Among these indicators are found those concerning job creation (E_DC, E_TC) and those related to the effort made by governments and individuals in favour of tourism development (GI and CI). This cluster can be associated with general good tourism development due to values higher than the mean of the region for almost all the indicators under consideration, with the exception of BTTS, DTTS, GI, CI, and OTS.

The fourth cluster is comprised of seven countries (Fig 3), all of which are island states. This cluster demonstrates an overall negative behaviour. Their positive performance was attained in the GI and CI indicators, as in the previous cluster, consistent with the general behaviour of the region. However, with the exception of these two indicators, four or more destinations show decreasing rates in all the indicators. This cluster can be associated with generally bad tourism development. The values of the indicator considered have been worsening on average throughout the period, and the most concerning issue is related to the number of jobs created by the tourism sector. Guadeloupe and Saint Vincent and the Grenadines only account for one and two positive slopes, respectively. The destinations of this cluster have an average score higher than the mean of the sample for the GI and OTS indicators.

The fifth cluster is the largest, with 17 countries (Fig 4), from which the majority are continental destinations (10). There are only seven island states in this cluster. The behaviour of this cluster is apparently not as homogeneous as the other clusters, where the changes in the variables are in the same direction. For the third and fourth clusters (Figs 2 and 4) most of the slopes are below or above zero, respectively.

A detailed inspection of Fig 4 reveals that 99% of the slopes are between -0.36 and 0.38, despite the seemingly erratic behaviour. On a more reduced scale, these values would be observed as being around zero. This is the cluster that registered the smallest average difference between the maximum and minimum slopes. The positive and negative slopes were similar in number for almost all the indicators with the exception of the GI with a single negative value, which is consistent with the regional view. Only 35% of the slopes were positive, which is also similar to the overall behaviour for Outbound Expenditure (OTS). The average performance for the countries of the cluster is lower than for the region. They only have an average score higher than the mean of the sample for the DTTS. As could be observed, despite the reduction of the information, it is impossible to attain a ranking for the destinations.

## Aggregative approach

In order to obtain a ranking, four composite measures are proposed. The first is called Posit-Slope and is obtained as the number of indicators for which destinations achieved positive slopes. In the case of a tie, the composite measurement would be the sum of the positive values. If the tie still remains, then, the values of the negative slopes will be subtracted from the sum of the positive slopes.

The second proposal involves the Mean of the slopes for each destination, and the third composite index is the Median thereof. These are easy-to-obtain descriptive values. The Mean is a compensatory methodology since it admits compensation between positive and negative scores in such a way that negative behaviour affects the destination scores, which include all the indicators: this is a good characteristic for composite indices. However, the method is influenced by extreme values. The Median is the most robust measure because it is not influenced by extreme values in that it only considers the values located in the middle of the distribution.

The fourth method is the Restrictive indicator. This is calculated as the sum of the values for the positive slopes of a destination, if it has no negative values. In the case of the presence of negative slopes, then the value of the Restrictive indicator is the sum of these negative values. This is a non-compensatory method, since it does not admit the presence of both positive and negative slopes in a destination.

Pearson's correlation coefficient (Table 3) was calculated for these values, as was Spearman's Rho for the rankings (Table 4). Each pair of methods attained coefficient values that were statistically significant at 0.01. As a result, it is possible to affirm that the rankings are statistically similar. The ranking corresponding to the restrictive method (Restrictive) appears in the last column of Table 1 (Ranking). These rankings were not compared to others, such as the TTCI, because they were obtained using the information from a time span. In contrast, other studies, [5,6,9,10,21] use static approaches to measure TDC. However, it is possible to observe that small destinations from the region have a higher level of competitiveness than those that have been established as being among the most competitive worldwide for a long time. This is the case of Aruba and Jamaica which achieved higher growth rates than Mexico, Colombia, Costa Rica, and Panama, just to mention a few of the destinations with better competitiveness positions from the static point of view [8].

Moreover, most of the destinations located among the 50% most competitive are those with greater mean growth in the indicators GDP_DC and GDP_TC. This is consistent with the findings of [16] that a rise in competitiveness brings about more than just a proportionate rise in the GDP.

A paired comparison of the rankings reveals that the greatest number of destinations that maintain the same positions are located in the PositSlope and the Restrictive rankings. However, they have an average variation of 4.42 units, due to the wide variation of the British Virgin Islands and Colombia (17 and 15 positions, respectively). The British Virgin Islands is worse in the Restrictive index than in the PositSlope, since, despite having just two negative slopes, they are the highest values for these two indicators in the sample. Therefore, this destination attained a low composite value with the Restrictive approach. On the other hand, Colombia improved 15 positions with the Restrictive method. This result arises since, even though this destination has only three indicators with positive behaviour in the period, the sum of the negative behaviours is lower than for other destinations, even those with a smaller number of negative slopes, such as the case of the British Virgin Islands. As a consequence, it is important for the destinations to have no negative performances or, at least, to guarantee small values for the negative slopes.

As stated in the literature, no method has been established as the most suitable for the creation of indicators to measure TDC [12] However, each of the proposed methodologies has its own advantages and weaknesses. The PositSlope and the Restrictive methodologies involve a major effort for each destination due to their non-compensatory character. As a result, a destination with good achievements in most of the indicators may be negatively affected by poor

**Table 3. Pearson's correlation.**

|  | Posit_Slope | Mean | Median | Restrictive |
|---|---|---|---|---|
| PositSlope | 1 | .924** | .871** | .766** |
| Mean |  | 1 | .941** | .825** |
| Median |  |  | 1 | .772** |
| Restrictive |  |  |  | 1 |

** Correlation is significant at the 0.01 level (2-tailed).

**Table 4. Spearman's Rho correlation.**

| | | PositSlope | Mean | Median | Restrictive |
|---|---|---|---|---|---|
| Spearman's Rho | PositSlope | 1 | .862** | .899** | .783** |
| | Mean | | 1 | .933** | .790** |
| | Median | | | 1 | .713** |
| | Restrictive | | | | 1 |

** Correlation is significant at the 0.01 level (2-tailed).

behaviour in one of a few indicators. On the other hand, the Mean and the Median methodologies allow for compensation between good and bad behaviour in the indicators, which is a more realistic view, given the fact that not all the destinations attained a positive performance in all the indicators in a given time span. These two may be affected by both internal and external factors. Notwithstanding, the proximity attained in the results verifies that all these methodologies are valid for the measurement of TDC. The choice of a method depends on the researcher/policy-maker decisions.

## Conclusions

This research comprises more indicators than were considered in previous investigations developed to analyse the competitiveness over a time span. The proposed indicators are representative of the elements that enable a diagnosis of the competitiveness situation of the destinations to be established, such as the determinants of TDC (GI, CI), the performance (DTS, LTTS, BTTS, VE, ITTC) and the impact (GDP_DC, GDP_TC, EDC, ETC), as stated in [13].

Furthermore, the study involves more destinations from the Caribbean region than there are in the TTCI [6,21], which constitutes a major contribution for studies in the area. Most of these destinations are considered to belong amongst the most tourism-dependent countries worldwide and have not previously been included in studies aimed to measure TDC. Moreover, other destinations, such as Guyana, Haiti, Puerto Rico, and Suriname, which had previously been excluded from the TTCI in the editions from 2017 and 2019, were also included, due to the importance of tourism for their economic development. It was possible to determine that several of these countries had improved the values of the indicators analysed, on average, by a greater margin than other countries considered competitive in global international rankings. Moreover, the analysis facilitated the consideration of all the available information in a given time span.

Methodologically, the proposed method enables a great number of destinations and years to be considered. It is easy to develop, the results are comprehensible, and it successfully demonstrates another way to analyse TDC, based on the slope of the regression equation for the indicator and the destination. Their values indicate the behaviour, either positive or negative, for each destination in a given period, and are not influenced by their size or level of tourism development. In addition, this method contributes towards the extension of the practical applications of Linear Regression techniques in the field of travel and tourism studies, and also recognises the signalling of [14] regarding the suitability of these techniques as essential tools for tourism management studies and their robustness when working with high volumes of hard data.

Destinations were defined as whole countries for this study due to the higher probability of obtaining accurate values for the indicators. Hard data or objective indicators were used because of their availability and for the possibility of future access over long periods in all the destinations evaluated. The innovation of this method consists of the use of the slopes of the regression equation calculated for each destination and indicator as being representative of the

destination's competitiveness in the period under study. This research has shed light on the way to analyse TDC while considering the number of destinations, indicators, and years covered.

The advantages of the study involve the possibility of comparing the competitiveness of a destination in a given period without the need to carry out calculations for each individual year. In general, the highest and lowest slope values are located in small island destinations with less participation in global ranking publications owing to incomplete information. Furthermore, the best behaviour corresponds to destinations with a lower level of economic development. The study also enabled the comparison of tourism-dependent destinations without taking into account their size or the stages of their tourism development. As a result, this method is applicable to all destinations without bearing in mind their size. The level of improvement of a destination over a period was employed to determine its ability to attain its best competitiveness position.

Cluster analysis revealed five clearly recognisable groups. Moreover, it was possible to identify a common pattern in the region through the indicators signalled by the Kruskal-Wallis test, such as those that cause no significant difference between groups. Among these featured the common interest of local governments to contribute towards the development of the tourist sector and the great importance given to the creation of tourist infrastructures and the support of services directly related to tourism development. Spending on domestic trips within the same territory has discreetly augmented, but less so than in other regions, which is consistent with the WTTC (2018b). There is similar behaviour regarding trips for business purposes to the countries of the area, which is lower with respect to other geographical regions, and negative behaviour of Outbound Expenditure, due to the economic conditions of the region.

A comparison between island and continental states was carried out, and their differences and similarities regarding their competitiveness were revealed. As a result, it was possible to demonstrate that island states are more tourism dependent than continental states. In general, it was possible to analyse the behaviour of one of the most tourist-intense regions worldwide with a detailed analysis of its countries. This research contributes towards solving the paradox of TDC and to the wide group of initiatives for its analysis. The study respects the relative nature of the TDC concept because the scores involve indicator information throughout the time span analysed. Furthermore, slope values explained the degree to which each destination annually improved or worsened in each indicator.

The results were consistent with the latest publications of the WTTC, for which the Caribbean region ranked first in issues concerning TDC, in relative terms, worldwide [43,51]. Moreover, the results enabled the countries that most influenced this global behaviour in the area to be identified. In general, this research proposes another way to analyse TDC and it was possible to prove that the measurement of destination competitiveness is not an easy task. The analysis was carried out using both non-aggregative and aggregative approaches.

For this final objective, four aggregation methods were proposed for the creation of global rankings. All the proposals have their own pros and cons. However, all were valid in that they received close results. Bearing this in mind, future research should be based on the creation of composite indicators of a more robust nature, while considering the methodological approaches and steps cited in the literature. This research is significant for decision-makers in determining the aspects that must be addressed to improve competitiveness, since it provides them with the opportunity of accessing a more detailed visualisation of the most influential factors, positively or negatively, of the destination's competitiveness and, therefore, of supporting the development of initiatives or policies that improve tourist development. Moreover, a greater number of indicators and years could be included in the process to increase the accuracy and time span analysed.

From the practitioner's standpoint, several implications emerge from our findings. First, it is possible to determine the competitiveness of the destinations in a time span and, therefore, to discover whether the policies and the decisions made enable the proposed objectives to be attained, which are measured by the tendencies of the values of the indicators over time. Moreover, the clusters permit the identification of the main competitors for each destination as those with a similar performance. The remaining clusters can be viewed as either positive or negative benchmarks. Consequently, the decision-making process should be encouraged with the determination of those indicators that require improvements to move from one cluster to another with better overall competitiveness. As a result, managers can now triangulate the data, which is extremely useful in evaluating which indicators affect the position of the destinations within each cluster, similar to the proposal of [52] and, consequently, serve as a guide for tourism planners, developers, and policy decision-makers as supported by [14].

One limitation of the study consists of the absence of external information to determine the weights or the relative importance of the indicators for their comparison with the rankings attained. Another limitation involves the exclusion of other small islands from the region due to the absence of available data. Furthermore, there is a lack of target values for each indicator and destination in the period, which would otherwise provide competitiveness of a more realistic nature, in that each destination would be evaluated in terms of the attainment of its own goals in the time span, while respecting the level of development and the internal planification of each destination.

## Supporting information

**S1 Appendix.**
(TIF)

## Author Contributions

**Conceptualization:** Víctor Ernesto Pérez León.

**Data curation:** Víctor Ernesto Pérez León, Maria Amparo León Sánchez.

**Formal analysis:** Flor Mª Guerrero.

**Funding acquisition:** Víctor Ernesto Pérez León, Maria Amparo León Sánchez, Flor Mª Guerrero.

**Investigation:** Víctor Ernesto Pérez León, Maria Amparo León Sánchez, Flor Mª Guerrero.

**Methodology:** Víctor Ernesto Pérez León, Maria Amparo León Sánchez.

**Supervision:** Flor Mª Guerrero.

**Visualization:** Víctor Ernesto Pérez León.

**Writing – original draft:** Víctor Ernesto Pérez León, Maria Amparo León Sánchez.

**Writing – review & editing:** Víctor Ernesto Pérez León.

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
