## [Decision Letter · Decision Letter 0]

13 Jan 2021

PONE-D-20-32669

Central American and Caribbean tourism destination’s competitiveness based on performance. A new approach

PLOS ONE

Dear Dr. Pérez León,

Thank you for submitting your manuscript to PLOS ONE. After careful consideration, we feel that it has merit but does not fully meet PLOS ONE’s publication criteria as it currently stands. Therefore, we invite you to submit a revised version of the manuscript that addresses the points raised during the review process.

- The authors should better define motivations for the research. Why do we need this study? What are advantage of this proposed methodology?

- Clearly define limitations of this study?

- Literature review section should be revised. Remove old references published before 2017. Clearly define a gap and how this study is filling it up.

- Discussion should be better organized with literature comparisons.

- Conclusions should be better organized.

We look forward to receiving your revised manuscript.

Kind regards,

Dragan Pamucar

Academic Editor

PLOS ONE

Journal Requirements:

"Authors would like to acknowledge the scholarship from the Asociación Universitaria Iberoamericana de Postgrado (AUIP)."

"L. M. A. Grant Number: PI-195, University of Pinar Del Río, Cuba https://www.upr.edu.cu

P. V. E. Grant Numbers: P18-RT-1566, UMA18-FEDERJA-065, Agencia de Innovación y Desarrollo de Andalucía

F. G. Grant Number: SEJ332. Secretaría General de Universidades, Investigación y Tecnología, Junta de Andalucía"

4. Please amend your manuscript to include your abstract after the title page.

5. We note that Figure 2 in your submission contain map images which may be copyrighted. All PLOS content is published under the Creative Commons Attribution License (CC BY 4.0), which means that the manuscript, images, and Supporting Information files will be freely available online, and any third party is permitted to access, download, copy, distribute, and use these materials in any way, even commercially, with proper attribution. For these reasons, we cannot publish previously copyrighted maps or satellite images created using proprietary data, such as Google software (Google Maps, Street View, and Earth). For more information, see our copyright guidelines: http://journals.plos.org/plosone/s/licenses-and-copyright.

5.1.    You may seek permission from the original copyright holder of Figure 2 to publish the content specifically under the CC BY 4.0 license. 

5.2.    If you are unable to obtain permission from the original copyright holder to publish these figures under the CC BY 4.0 license or if the copyright holder’s requirements are incompatible with the CC BY 4.0 license, please either i) remove the figure or ii) supply a replacement figure that complies with the CC BY 4.0 license. Please check copyright information on all replacement figures and update the figure caption with source information. If applicable, please specify in the figure caption text when a figure is similar but not identical to the original image and is therefore for illustrative purposes only.

6. Please include a caption for figure 2.

Reviewers' comments:

Reviewer's Responses to Questions

**Comments to the Author**

1. Is the manuscript technically sound, and do the data support the conclusions?

Reviewer #1: Yes

2. Has the statistical analysis been performed appropriately and rigorously? 

Reviewer #1: Yes

3. Have the authors made all data underlying the findings in their manuscript fully available?

Reviewer #1: Yes

4. Is the manuscript presented in an intelligible fashion and written in standard English?

Reviewer #1: No

5. Review Comments to the Author

Reviewer #1: The work is mainly an application to a broad set of countries of the model of Dwyer and Kim (2003) and Ritchie and Crouch. The paper is well structured and rigorous from an empirical point of view, however it has some major issues in order to be published. Refering to the literature, destination competitiveness is both a determining variable of tourism performance and a partial outcome of performance - it is inherently linked without there necessarily being one-way causality. Need better justification for this issue.

My first concern regards to the definition of destinations and the tourism destination competitiveness model. In the original idea of Ritchie and Crouch and Dwyer and Kim, destinations are usually defined as tourist destination, while in the paper seems that destination are defined as countries. The author(s) introduces several conceptual models on tourism destination competitiveness but this review in not well related to the following empirical model. There is no economic or methodological discussion justifying the variables selection in your study model.

In the analysis, a large set of variables from TTCI is used to measure TDC and to verify structural relationships among TCD, TTCI and performance. I am not completely convinced this is a robust procedure. The set of variables from TTCI, used to measure TCD, is obviously highly correlated with the TTCI ranking, which is calculated as the mean of (a subset of) these items. Next, a lot of countries were excluded. Even the authors explain why they excluded countries, the issues remain: May this exclusion affect model estimates?.

There are a number of serious difficulties with this paper - especially the way it was written. What are the main and relevant findings of the analysis? What is the economic and managerial implications of the model estimates? How empirical results are relevant in suggesting the correct method to measure competitiveness between destinations? All these aspects should be better discussed.

6. PLOS authors have the option to publish the peer review history of their article (what does this mean?). If published, this will include your full peer review and any attached files.

Reviewer #1: **Yes: **Mohd Hafiz Hanafiah

---

## [Author Response · Author response to Decision Letter 0]

23 Apr 2021

Dear Editors and Reviewers:

First of all, we would like to acknowledge your time and efforts in the reviewing of the paper and give you thanks for the comments and suggestions. Your recommendations have certainly contributed to the improvement of our research and to the quality of the document.

We hope that the changes made fulfill the quality requirements for our paper to be published in the Journal Plos One.

The Editors (Ed.) and reviewers’ comments (Rev.) appear in black colour. Our answers are letter red colour. The new text included in the document as well as the information modified is also appear in red colour and italics.

Ed.1 - The authors should better define motivations for the research. Why do we need this study?:

The authors coincide with the editors in this comment. To fulfil this requirement come changes were made in the document and the Introduction section was the rewritten. The motivations of the study have been clearly stated in the document as follows:

Line 48 to 86: “and despite the efforts registered in the literature, the results thereof measurement of destination competitiveness remain tenuous (7). However, the progress presented to date reveals several limitations regarding the selection of evaluation variables, the calculation of their respective weights (8), and the aggregation procedure used. Furthermore, we would like to point out that the temporal aspect constitutes a further issue regarding the evaluation of TDC.

Diverse studies evaluate TDC at a given moment in time (9–11), which is a widely used and convenient approach. This enables the competitive position of a destination with respect to its main competitors to be assessed at a certain moment, whereby the competitiveness is evaluated from a static point of view. As a result, the competitive position of a destination is evaluated by considering the values of the indicators at a given moment and, therefore, it is impossible to evaluate whether the policies and decisions made have contributed towards the improvement of the competitiveness along time.

Under this approach, the level of competitiveness of a certain destination may be affected by external factors in such a way that other destinations could be considered more competitive, not from improving the values of their indicators, but instead due to the deterioration of this first destination. In this respect, the use of a dynamic approach, which permits competitiveness to be analysed over a period of time, enables a destination to be analysed as to whether it is capable of improving its levels of competitiveness over time, in such a way that a higher value of competitiveness is associated to increasing behaviour over time.

Within the static approach, a latent debate rages on regarding which is the most feasible methodology to create TDC rankings (12). However, despite advances in the area (9–11), differences between the proposed procedures remain, and some of the proposed alternatives imply the utilisation of algorithms that can, on occasions, create measures of tourism competitiveness that are difficult to explain to decision-makers. This seriously reduces their usefulness to end users.

Other researchers evaluated the TDC within a time span (13–16). Most of these studies are more focused on the determination of the factors that influence the competitiveness of the destinations than on the analysis of the competitiveness positions of the destinations. Some researchers suggest segmenting the sample based on the destination characteristics and then analysing these segments based on smaller sub-samples of similar destinations (14). However, these proposals continue to seek a better way to employ all the information. In this respect, the present research aims to contribute towards the literature in seeking methods to measure TDC by addressing several of the aforementioned gaps in research. First, the proposal involves measuring the TDC over a period of time, thereby analysing it from a dynamic approach. 

In order to make a suitable diagnosis of competitiveness in this sector, it is important in the analysis to take into consideration the period in which impacts occur (13).” 

Ed. 1.1. What are advantage of this proposed methodology?

The advantage of the proposed methodology has been inserted in different parts of the document as follows:

 Lines from 94 to 103: “As a result, it will be possible to analyse whether the administrative decisions made over time contribute towards the improvement of the behaviour of the destinations, measured by the average behaviour of the indicator values.

The proposal determines the way in which destinations improve their TDC during a time span with respect to themselves (1); that is, whether each destination manages to achieve, to a greater or lesser extent, its objectives, in accordance with its development possibilities and economic conditions. Moreover, the study allows the way in which the destinations improve their TDC with respect to their competitors to be investigated (10,17), thereby taking into account the importance that comparison with other competitors holds in the measurement of TDC”

Lines from 105 to 117: “A new way to analyse the initial information is presented. The proposed method is easy to apply, and the results are comprehensible and, therefore, easy to interpret for the end users. Moreover, the results can be analysed using either the non-aggregative or the aggregative approach. This latter approach involves the creation of a composite index. To this end, various methods are also proposed to take advantage of the information covering the whole time span.

This allows us to investigate a way to measure competitiveness over a given period of time, in such a way that it is seen as a dynamic and not a static process, whose results are easily understood by decision-makers. This process enables the behaviour of the level of competitiveness of destinations to be analysed visually. Furthermore, the proposal permits competitiveness to be analysed over a period of time without the need to carry out the calculation procedure for each year or sub-period analysed, thereby reducing the computational cost.”

Ed.2. Clearly define limitations of this study?

The limitations of the study were inserted at the end of the conclusions’ section, from line 883 to 890.

“One limitation of the study consists of the absence of external information to determine the weights or the relative importance of the indicators for their comparison with the rankings attained. Another limitation involves the exclusion of other small islands from the region due to the absence of available data. Furthermore, there is a lack of target values for each indicator and destination in the period, which would otherwise provide competitiveness of a more realistic nature, in that each destination would be evaluated in terms of the attainment of its own goals in the time span, while respecting the level of development and the internal planification of each destination”

Ed.3. - Literature review section should be revised. Remove old references published before 2017. Clearly define a gap and how this study is filling it up.

The Literature review section was revised. The old references were eliminated, and new references have been added as can be shown in the References section. (The new text is not included in the rebuttal letter because its extensiveness The new text added to the paper is not included in the rebuttal letter because its great extensiveness).

 In addition, the definition of the gap and the way in which this study is filling has been mentioned in this document in the previous answers: Ed.1 and Ed. 1.1. Moreover, From lines 122 to 125 it is said:

“a further innovation of the study is that of the sample. This is a unique dataset of 33 destinations from the Caribbean region, which is, to the best of our knowledge, one of the largest samples of this kind in the studies developed in the region that involves destinations and number of years”

Ed.4. - Discussion should be better organized with literature comparisons.

To make the Discussion more comprehensible, new information was added and also the comparison with literature was included following the editor’s suggestions:

Lines from 495 to 500: “The study of competitiveness is to be carried out based on the slope of the regression equation for each indicator and destination. The growth rates of the indicators are more approximate to the concept of competitiveness because they indicate the change in the time of those acquired levels (13). The analysis will be made using both the non-aggregative and the aggregative approaches.”

Lines from 526 to 537: “Under this approach, if a destination performs equal to or better than that expected, then a positive competitiveness diagnosis is established, and a negative competitiveness diagnosis is established if the flow performs worse than predicted, as is stated by (8). As a result, positive values demonstrate that a given destination maintains improvement towards a more competitive position with respect to its own planification. A growing tendency reveals the positive impact of the policies and decisions made whose objective is to develop the tourism sector. Following this explanation, a destination with a slope greater than others in a certain indicator demonstrates improved competitiveness. Consequently, it is possible to evaluate the level of competitiveness of a destination with respect to itself and with respect to the other competitors.

The analysis can be developed by means of indicator, to identify regional trends or by destination, to evaluate the attainment of individual competitiveness”

Lines 567 and 568: “The analysis may be also carried out under the destination approach, to analyse individual behaviour”

Lines from 614 to 617: “It is therefore possible to identify how small destinations such as Aruba have attained higher growth rates than others that have been established as competitive destinations in the international tourist market for a long time (8).”

Lines from 647 to 650: “consistent with the achievement of (20) in the measurement of TDC. Moreover, this is also associated to the outputs of (14) for the tourism-based countries that recognise the importance of developing resources that generate value for the tourism industry and for the broader economy.”

Lines from 654 to 658: “This is also related to the findings of (20) reveal the close relationship of the variables that measure tourism spending and competitiveness. In addition, the region is located in one of the last positions worldwide for these two items according to (51) which demonstrate low regional competitiveness”

Lines from 742 to 754: “These rankings were not compared to others, such as the TTCI, because they were obtained using the information from a time span. In contrast, other studies, (5,6,9,10,21) use static approaches to measure TDC. However, it is possible to observe that small destinations from the region have a higher level of competitiveness than those that have been established as being among the most competitive worldwide for a long time. This is the case of Aruba and Jamaica which achieved higher growth rates than Mexico, Colombia, Costa Rica, and Panama, just to mention a few of the destinations with better competitiveness positions from the static point of view (8).

Ed.4 Moreover, most of the destinations located among the 50% most competitive are those with greater mean growth in the indicators GDP_DC and GDP_TC. This is consistent with the findings of (16) that a rise in competitiveness brings about more than just a proportionate rise in the GDP.”

Ed. 4.1- Conclusions should be better organized.

The conclusions also were organized following this comment. The changes made are located in:

Lines from 786 to 790: “The proposed indicators are representative of the elements that enable a diagnosis of the competitiveness situation of the destinations to be established, such as the determinants of TDC (GI, CI), the performance (DTS, LTTS, BTTS, VE, ITTC) and the impact (GDP_DC, GDP_TC, EDC, ETC), as stated in (13).”

Lines from 807 to 811: “In addition, this method contributes towards the extension of the practical applications of Linear Regression techniques in the field of travel and tourism studies, and also recognises the signalling of (14) regarding the suitability of these techniques as essential tools for tourism management studies and their robustness when working with high volumes of hard data”

Lines from 832 to 835: “Among these featured the common interest of local governments to contribute towards the development of the tourist sector and the great importance given to the creation of tourist infrastructures and the support of services directly related to tourism development.”

Moreover, the implications for partitioners and the limitations of the study were added to this section from lines 868 to 888, coincident to previous comments ¡made by the Editors and the Reviewers.

"Authors would like to acknowledge the scholarship from the Asociación Universitaria Iberoamericana de Postgrado (AUIP)."

This issue has been solved and the funding information does not appear in the Acknowledgement section. The funding information corresponding to the “Asociación Universitaria Iberoamericana de Postgrado (AUIP)” will be included correctly in the Funding Statement.

"L. M. A. Grant Number: PI-195, University of Pinar Del Río, Cuba https://www.upr.edu.cu

P. V. E. Grant Numbers: P18-RT-1566, UMA18-FEDERJA-065, Agencia de Innovación y Desarrollo de Andalucía.

F. G. Grant Number: SEJ332. Secretaría General de Universidades, Investigación y Tecnología, Junta de Andalucía"

I would ask in the cover letter to include the following funding information:

“P. V. E. Grant Number: RES/19/12/2017, Asociación Universitaria Iberoamericana de Postgrado (AUIP).”

 4. Please amend your manuscript to include your abstract after the title page.

The Abstract has been included after the title page

 5. We note that Figure 2 in your submission contain map images which may be copyrighted. All PLOS content is published under the Creative Commons Attribution License (CC BY 4.0), which means that the manuscript, images, and Supporting Information files will be freely available online, and any third party is permitted to access, download, copy, distribute, and use these materials in any way, even commercially, with proper attribution. For these reasons, we cannot publish previously copyrighted maps or satellite images created using proprietary data, such as Google software (Google Maps, Street View, and Earth). For more information, see our copyright guidelines: http://journals.plos.org/plosone/s/licenses-and-copyright.

Figure 2 has been removed from my submission, due to the unavailability of permission from the copyright holder.

Reviewers' comments:

Reviewer's Responses to Questions

Comments to the Author

1. Is the manuscript technically sound, and do the data support the conclusions?

Reviewer #1: Yes

2. Has the statistical analysis been performed appropriately and rigorously? 

Reviewer #1: Yes

3. Have the authors made all data underlying the findings in their manuscript fully available?

Reviewer #1: Yes

4. Is the manuscript presented in an intelligible fashion and written in standard English?

Reviewer #1: No

We would like to thanks to the reviewer for this observation. To solve this problem, the paper has been reviewed completely. Some changes have been made in the redaction of the document (coloured in red) and the language has been revised by an English native speaker. 

5. Review Comments to the Author

Reviewer #1: The work is mainly an application to a broad set of countries of the model of Dwyer and Kim (2003) and Ritchie and Crouch. The paper is well structured and rigorous from an empirical point of view, however it has some major issues in order to be published. Refering to the literature, destination competitiveness is both a determining variable of tourism performance and a partial outcome of performance - it is inherently linked without there necessarily being one-way causality. Need better justification for this issue.

Rev.1. My first concern regards to the definition of destinations and the tourism destination competitiveness model. In the original idea of Ritchie and Crouch and Dwyer and Kim, destinations are usually defined as tourist destination, while in the paper seems that destination are defined as countries. 

The authors would like to thank the reviewer for this comment. Indeed, countries are the destinations in this paper. To justify this choice, the following information has been added to the document with the corresponding references, from line 119 to 123:

“In this study, the destinations are the countries, whereby their facility to provide the information is considered. This helps towards the better provision of data when diverse years are included in the analysis. As a consequence, a further innovation of the study is that of the sample. This is a unique dataset of 33 destinations from the Caribbean region…”

Moreover, from lines 190 to 198: 

“This has been addressed at several levels (i.e., firm, regional, and national levels) (7); and has included resorts (24) regional locations within the same country (8,27,29,30) or destinations from different countries (3), tour operator and hotel companies (31), cities (32), municipalities (2), regions (1,26) and countries (4,6,10,13,20). Given the aim of the present research to include countries from the Caribbean region in TDC studies and also to consider the aforementioned availability of information, the selected destination size for the study is that of the country level. This is consistent with the view of several of the aforementioned applications.” 

Rev. 1.1 The author(s) introduces several conceptual models on tourism destination competitiveness but this review in not well related to the following empirical model.

The authors agreed with the reviewer in this sense. To avoid this issue, the reference to the conceptual models has been erased considering that the main aim of the paper is to contribute to the measurement of TDC in a given time span. Therefore, the title has also be changed.

The initial tittle was: “Central American and Caribbean tourism destination’s competitiveness based on performance. A new approach” 

The new tittle proposed: “Central American and Caribbean tourism destinations’ competitiveness: A temporal approach”

Rev. 1.2. There is no economic or methodological discussion justifying the variables selection in your study model.

The justification of the variables used in the study has been included in the text (Lines from 400 to 427) as follows, revealing their relationship to TDC and their previous use in studies aimed to measure it. 

“Given the potential importance and contribution of tourism to a country's GDP, and of its benefits to a wide range of economic activities in the context of increased global competition, tourist destinations have been forced to seek new ways to obtain a competitive advantage (6). Indicators GDP_DC and GDP_TC quantify the relative importance of the tourism industry in each destination, and are valid for measurement of TDC (44). According to (17), an improvement in Travel & Tourism competitiveness is an encouraging trend given that, in over half the countries in the Americas, the Travel & Tourism industry’s share of GDP is greater than the aggregate global level. Those indicators that consider employment in the sector (EDC and ETC) are of major importance since competitive destinations provide and increase employment and value added by the tourism industry (45). These indicators have been used in dimensions aimed at monitoring the evolution of destination competitiveness (20).

The development of tourism contributes positively towards the economic prosperity of countries for which the bidirectional causal relationship can be emphasised (20,46). Tourism spending is undoubtedly a major key factor not only in terms of economic growth but also in terms of competitiveness (6). The indicators concerning tourism spending therefore remain useful in measuring TDC. This is the case of DTTS, LTTS, BTTS, and ITTC used by (20) to evaluate competitiveness, and of the indicator VE, also used by (17) to analyse destination competitiveness, and of Outbound Tourism (OTS) (7). 

The extent to which the government prioritises the Travel and Tourism sector exerts a significant impact on its competitiveness (6) In this respect, national and local governments play vital roles in tourism development (5,6). The indicator GI is representative of the efforts of governments to encourage tourism development. From among the indicators used in the study, this is the only indicator that coincides with those proposed by the WEF to create the TTCI (20). This indicator has also been used in other studies aimed at measuring TDC (16). Moreover, Capital Investment (CI) may be viewed as a contribution to local economies to invest in the tourism sector, whose objective is to lead to economic benefits.”

Rev. 2. In the analysis, a large set of variables from TTCI is used to measure TDC and to verify structural relationships among TCD, TTCI and performance. I am not completely convinced this is a robust procedure. The set of variables from TTCI, used to measure TCD, is obviously highly correlated with the TTCI ranking, which is calculated as the mean of (a subset of) these items. 

The authors would like to thanks to the reviewer for this comment. In this regard, we would like to point out that the variables used in our study are not included in the TTCI. These variables were gathered from the World Travel and Tourism Council. To clarify this issue the following information has been added in the document:

Lines from 422 to 424: “From among the indicators used in the study, this is the only indicator that coincides with those proposed by the WEF to create the TTCI (20).”

Moreover, to avoid misunderstandings, the authors have decided to eliminate the analysis of performance, following the suggestion of the reviewers, agreeing that “destination competitiveness is both a determining variable of tourism performance and a partial outcome of performance, it is inherently linked without there necessarily being one-way causality” as they stated. Therefore, the aim of the research is as follows:

Lines from 81 to 84: “In this respect, the present research aims to contribute towards the literature in seeking methods to measure TDC by addressing several of the aforementioned gaps in research. First, the proposal involves measuring the TDC over a period of time, thereby analysing it from a dynamic approach.”

With regard to the comparison between each pair of rankings, it should be noted that giving the fact that the proposal comprises a time span, no comparison was made with the TTCI. The TTCI proposes a static approach, and the proposal of this paper involves a dynamic method which embraces all the information from 2000 to 2019, starting seven years before the first edition of the TTCI. Notwithstanding, the following explanation was included in the document from line 742 to 750: 

“These rankings were not compared to others, such as the TTCI, because they were obtained using the information from a time span. In contrast, other studies, (5,6,9,10,21) use static approaches to measure TDC. However, it is possible to observe that small destinations from the region have a higher level of competitiveness than those that have been established as being among the most competitive worldwide for a long time. This is the case of Aruba and Jamaica which achieved higher growth rates than Mexico, Colombia, Costa Rica, and Panama, just to mention a few of the destinations with better competitiveness positions from the static point of view (8).”.

Rev. 2.1 Next, a lot of countries were excluded. Even the authors explain why they excluded countries, the issues remain: May this exclusion affect model estimates?

The authors would like to thanks to the reviewers for this comment. Considering the scope of our study, it is more an inclusion of destinations than an omission. The following explanation was added to justify the number of destinations used.

Lines from 212 to 125: “As a consequence, a further innovation of the study is that of the sample. This is a unique dataset of 33 destinations from the Caribbean region, which is, to the best of our knowledge, one of the largest samples of this kind in the studies developed in the region that involves destinations and number of years.”

Lines from 132 to 134: “Moreover, only 17 from the 33 destinations incorporated in the study have been considered at least once in the Travel and Tourism Competitiveness Index of the World Economic Forum (6)…”

Lines from 140 to 143: “…such as various destinations in the Caribbean region, which have been excluded from the editions from 2017 and 2019 due to information unavailability, despite being the most tourism-dependent region worldwide (18).”

Lines from 795 to 797: “Moreover, other destinations, such as Guyana, Haiti, Puerto Rico, and Suriname, which had previously been excluded from the TTCI in the editions from 2017 and 2019, were also included, due to the importance of tourism for their economic development.”

Rev. 3. There are a number of serious difficulties with this paper - especially the way it was written. 

Thanks to the comments by both, the reviewers and the Editors, the document has been modified and we consider that this is an improved version

Rev. 3.1. What are the main and relevant findings of the analysis? 

The findings of the analysis have been inserted in the Results and Conclusions sections as follows:

Lines from 615 to 617: “It is therefore possible to identify how small destinations such as Aruba have attained higher growth rates than others that have been established as competitive destinations in the international tourist market for a long time (8).”

Lines from 751 to 754: “Moreover, most of the destinations located among the 50% most competitive are those with greater mean growth in the indicators GDP_DC and GDP_TC. This is consistent with the findings of (16) that a rise in competitiveness brings about more than just a proportionate rise in the GDP.”

Lines from 786 to 790: “The proposed indicators are representative of the elements that enable a diagnosis of the competitiveness situation of the destinations to be established, such as the determinants of TDC (GI, CI), the performance (DTS, LTTS, BTTS, VE, ITTC) and the impact (GDP_DC, GDP_TC, EDC, ETC), as stated in (13).”

Lines from 797 to 801: “It was possible to determine that several of these countries had improved the values of the indicators analysed, on average, by a greater margin than other countries considered competitive in global international rankings. Moreover, the analysis facilitated the consideration of all the available information in a given time span.”

Lines from 807 to 8011: “In addition, this method contributes towards the extension of the practical applications of Linear Regression techniques in the field of travel and tourism studies, and also recognises the signalling of (14) regarding the suitability of these techniques as essential tools for tourism management studies and their robustness when working with high volumes of hard data.”

Lines from 832 to 835: “Among these featured the common interest of local governments to contribute towards the development of the tourist sector and the great importance given to the creation of tourist infrastructures and the support of services directly related to tourism development.”

Rev. 3.2. What is the economic and managerial implications of the model estimates? 

The managerial implications have been added in the paper, from lines 868 to 880 as follows:

“From the practitioner’s standpoint, several implications emerge from our findings. First, it is possible to determine the competitiveness of the destinations in a time span and, therefore, to discover whether the policies and the decisions made enable the proposed objectives to be attained, which are measured by the tendencies of the values of the indicators over time. Moreover, the clusters permit the identification of the main competitors for each destination as those with a similar performance. The remaining clusters can be viewed as either positive or negative benchmarks. Consequently, the decision-making process should be encouraged with the determination of those indicators that require improvements to move from one cluster to another with better overall competitiveness. As a result, managers can now triangulate the data, which is extremely useful in evaluating which indicators affect the position of the destinations within each cluster, similar to the proposal of (52) and, consequently, serve as a guide for tourism planners, developers, and policy decision-makers as supported by (14).”

Rev. 3.3. How empirical results are relevant in suggesting the correct method to measure competitiveness between destinations? All these aspects should be better discussed.

This comment was addressed in results section, adding the following information from line 770 to 782:

“As stated in the literature, no method has been established as the most suitable for the creation of indicators to measure TDC (12) However, each of the proposed methodologies has its own advantages and weaknesses. The PositSlope and the Restrictive methodologies involve a major effort for each destination due to their non-compensatory character. As a result, a destination with good achievements in most of the indicators may be negatively affected by poor behaviour in one of a few indicators. On the other hand, the Mean and the Median methodologies allow for compensation between good and bad behaviour in the indicators, which is a more realistic view, given the fact that not all the destinations attained a positive performance in all the indicators in a given time span. These two may be affected by both internal and external factors. Notwithstanding, the proximity attained in the results verifies that all these methodologies are valid for the measurement of TDC. The choice of a method depends on the researcher/policy-maker decisions.”

 

Some other changes have been made in the document to answer different comments made by the reviewers:

1. What do you mean by this statement? Elaborate (Line 50). 

Answer: Lines from 47 to 51: “The measurement and comparison of tourism competitiveness is not an easy task, and despite the efforts registered in the literature, the results thereof measurement of destination competitiveness remain tenuous (7). However, the progress presented to date reveals several limitations regarding the selection of evaluation variables, the calculation of their respective weights (8)”

2. From 2011 studies. Many had change since then. Update your literature and avoid such strong statements. (Line 54).

Answer: The literature has been updated, as can be seen in the document. Additionally, a statement similar to the affirmation from 2011 was inserted from documents published in 2018 and 2020.

3. Are you talking about OLS and the Gravity model. (Line 62).

Answer: Lines from 86 to 88: “The proposal involves using the slope of the regression equation for each indicator in each destination, based on ordinary least squares (OLS)”

4. How about structural equation modelling? (Line 129)

Answer: It was also included and referenced in the text on line 182: “Moreover, Structural equation modelling should be mentioned (28)” 

5. Are we focusing on the competitiveness or performance as the DV? (Line 180)

Answer: The focus of the paper has been changed completely to competitiveness as was stated on the answer of comment Rev. 1.1.

6. So why the authors highlighted the use of soft data in your introduction section if you are not planning to use it?

Answer: The explanation of the use of soft data to evaluate TDC has been omitted, in order to avoid misunderstanding, following the suggestions of the reviewer. More attention was given to the use of hard data. Line 168 to 172: “Hard data, however, typically included in assessments of destination competitiveness, helps to conveniently gather large volumes of data and destinations (7,13,26), leads to more precise and accurate results, and is available over time, thereby facilitating the realisation of longitudinal studies. Hence, hard data is the type proposed in the present research”

7. Why no opt for log linear model – explain the relationship based on elasticities. (line 443)

To answer this question the following information was added from line 507 to 509: “No other models were selected, since the aim is to demonstrate the behaviour of the indicators over time, thereby guaranteeing the greater explanatory power of the results. To this end, the linear model is the most suitable.”

8. Any issue with multicollinearity? (Line 694).

Answer: Multicollinearity is not a problem because linear regression is calculated for a single indicator each time. However, a certain degree of correlation between the indicators is desirable since they are all selected to explain the same phenomenon.

---

## [Decision Letter · Decision Letter 1]

11 May 2021

Central American and Caribbean Tourism destinations' competitiveness: A temporal approach

PONE-D-20-32669R1

Dear Dr. Pérez León,

We’re pleased to inform you that your manuscript has been judged scientifically suitable for publication and will be formally accepted for publication once it meets all outstanding technical requirements.

Kind regards,

Dragan Pamucar

Academic Editor

PLOS ONE

Additional Editor Comments (optional):

Reviewers' comments:

Reviewer's Responses to Questions

**Comments to the Author**

1. If the authors have adequately addressed your comments raised in a previous round of review and you feel that this manuscript is now acceptable for publication, you may indicate that here to bypass the “Comments to the Author” section, enter your conflict of interest statement in the “Confidential to Editor” section, and submit your "Accept" recommendation.

Reviewer #1: All comments have been addressed

2. Is the manuscript technically sound, and do the data support the conclusions?

Reviewer #1: Yes

3. Has the statistical analysis been performed appropriately and rigorously? 

Reviewer #1: Yes

4. Have the authors made all data underlying the findings in their manuscript fully available?

Reviewer #1: Yes

5. Is the manuscript presented in an intelligible fashion and written in standard English?

Reviewer #1: Yes

6. Review Comments to the Author

Reviewer #1: Thank you for your extensive feedback and correction. The revised article had been improved tremendously. Therefore, I believe that the paper is ready for publication.

7. PLOS authors have the option to publish the peer review history of their article (what does this mean?). If published, this will include your full peer review and any attached files.

Reviewer #1: **Yes: **Mohd Hafiz Hanafiah

---

## [Editor Report · Acceptance letter]

17 May 2021

PONE-D-20-32669R1 

Central American and Caribbean tourism destinations’ competitiveness:  A temporal approach 

Dear Dr. Pérez León:

I'm pleased to inform you that your manuscript has been deemed suitable for publication in PLOS ONE. Congratulations! Your manuscript is now with our production department. 

Kind regards, 

on behalf of

Dr. Dragan Pamucar 

Academic Editor

PLOS ONE